# Intraoperative MRI Assessment of the Tissue Damage during Laser Ablation of Hypothalamic Hamartoma

**DOI:** 10.3390/diagnostics13142331

**Published:** 2023-07-10

**Authors:** Sophie Lombardi, Domenico Tortora, Stefania Picariello, Sniya Sudhakar, Enrico De Vita, Kshitij Mankad, Sophia Varadkar, Alessandro Consales, Lino Nobili, Jessica Cooper, Martin M. Tisdall, Felice D’Arco

**Affiliations:** 1Radiodiagnostic Department, Fondazione IRCCS Policlinico San Matteo, 27100 Pavia, Italy; sophielombardi@mac.com; 2Neuroradiology Unit, IRCCS Istituto Giannina Gaslini, 16147 Genoa, Italy; domenicotortora@gaslini.org; 3Neuro-Oncology Unit, Department of Paediatric Oncology, Santobono-Pausilipon Children’s Hospital, 80123 Naples, Italy; stefaniapicariello34@gmail.com; 4Great Ormond Street Hospital for Children NHS Foundation Trust, London WC1N 3JH, UK; sniya.sudhakar@gosh.nhs.uk (S.S.); enrico.devita@gosh.nhs.uk (E.D.V.); kshitij.mankad@gosh.nhs.uk (K.M.); jessica.cooper@gosh.nhs.uk (J.C.); martin.tisdall@gosh.nhs.uk (M.M.T.); 5Department of Surgical Sciences, Division of Neurosurgery, IRCCS Istituto Giannina Gaslini, 16147 Genoa, Italy; alessandroconsales@gaslini.org; 6Child Neuropsychiatry Unit, IRCCS Istituto Giannina Gaslini, 16147 Genoa, Italy; linonobili@gaslini.org

**Keywords:** gelastic epilepsy, hypothalamic hamartoma, MRgLITT, stereotactic laser ablation, intraoperative MRI, epilepsy surgery

## Abstract

Laser ablation for treatment of hypothalamic hamartoma (HH) is a minimally invasive and effective technique used to destroy hamartomatous tissue and disconnect it from the functioning brain. Currently, the gold standard to evaluate the amount of tissue being “burned” is the use of heat maps during the ablation procedure. However, these maps have low spatial resolution and can be misleading in terms of extension of the tissue damage. The aim of this study is to use different MRI sequences immediately after each laser ablation and correlate the extension of signal changes with the volume of malacic changes in a long-term follow-up scan. During the laser ablation procedure, we imaged the hypothalamic region with high-resolution axial diffusion-weighted images (DWI) and T2-weighted images (T2WI) after each ablation. At the end of the procedure, we also added a post-contrast T1-weighted image (T1WI) of the same region. We then correlated the product of the maximum diameters on axial showing signal changes (acute oedema on T2WI, DWI restriction rim, DWI hypointense core and post-contrast T1WI rim) with the product of the maximum diameters on axial T2WI of the malacic changes in the follow-up scan, both as a fraction of the total area of the hamartoma. The area of the hypointense core on DWI acquired immediately after the laser ablation statistically correlated better with the final area of encephalomalacia, while the T2WI, hyperintense oedema, DWI rim and T1WI rim of enhancement tended to overestimate the encephalomalacic damage. In conclusion, the use of intraoperative sequences (in particular DWI) during laser ablation can give surgeons valuable information in real time about the effective heating damage on the hamartomatous tissue, with better spatial resolution in comparison to the thermal maps.

## 1. Introduction

Hypothalamic hamartoma (HH) or tuber cinereum hamartoma is a rare, benign, non-neoplastic and non-progressive malformative congenital lesion, composed by normal mature neurons, glia and fiber bundles with an abnormal cytological architecture.

It occurs during fetal development, probably as a migration anomaly [1]. Most cases of HH are sporadic, but an association with Pallister–Hall syndrome has been described.

Clinical manifestations can vary and are related to location: posterior hypothalamus and mammillary bodies location is generally characterized by gelastic seizures without impairment of consciousness, typically with onset during the first year of life, while anterior hypothalamic or tuber cinereum location is associated with central precocious puberty, related to an increase of gonadotropin-releasing hormone [1].

Gelastic seizures often respond poorly to pharmacological therapy; 73% of patients have daily attacks and require surgical management. Furthermore, other types of seizures (tonic-clonic, focal, atonic and absence) can also be associated with HH. There is a spectrum of learning from normal development to progressive decline with significant intellectual disability. Neuropsychiatric disorders are common, including rage attacks, autism, attention deficit hyperactivity disorder, anxiety and phobia [2].

The surgical treatment of HH consists in the removal or disconnection of the hamartoma from the normal brain, interrupting the seizure network. Currently, besides open surgery, minimally invasive techniques are available, such as stereotactic radiosurgery (SRS), radiofrequency ablation (RFA) and laser interstitial thermal therapy (LITT) [2,3,4]. Stereotactic radiosurgery and radiofrequency ablation have some limitations compared to laser interstitial thermal therapy.

SRS has the main disadvantage of requiring up to 3 years to demonstrate the efficacy of the treatment and is also associated with a more significant risk of damaging radio-sensitive surrounding structures like the optic chiasm.

RFA allows for ablating a more limited area in a single session than is possible to do with LITT, and it also requires the use of multiple probes with different trajectories [4].

MRgLITT is a minimally invasive technique that was developed over the last decade, offering an alternative to open craniotomy, with shorter hospital stay, high rate of seizure control and reduced morbidity, becoming the first-choice procedure for HH treatment.

Compared with radiofrequency ablation and stereotactic radiosurgery, MRgLITT has the advantage of being performed with MRI compatible equipment, thus allowing for real-time assessment of the tissue ablation during the procedure [1,2,4].

It can be used as first-line treatment or as a second-line treatment for previously unsuccessfully treated lesions. MRI monitoring is performed using intraoperative MR-thermal maps that provide a real-time temperature monitoring during the procedure (every few seconds); however, these maps have a low spatial resolution compared to morphological MR sequences [5,6]. The possibility to integrate a higher spatial resolution intraoperative MR protocol of the target tissue may give the surgeon adjunctive information regarding the damage to the surrounding tissue, leading to more efficient treatment and reducing the need for further treatment.

## 2. Aim

The aim of the study was, first, to evaluate intraoperative signal changes in axial diffusion-weighted images (DWI) and T2-weighted images (T2WI) after each laser ablation and the area of the rim enhancement in post-contrast T1-weighted images (T1WI) acquired at the end of the treatment. Then, we aimed to see which of these better correlated with the final malacic damage within the hamartoma on follow-up images. In other words, our goal was to see which intraoperative sequence better predicts the final radiological damage (i.e., encephalomalacia).

## 3. Material and Methods

### 3.1. Study Population and Imaging Analysis

Initially, 39 patients who underwent laser ablation in two tertiary pediatric hospitals since January 2018 were studied. The exclusion criteria were as follows: (1) size of the hamartoma <5 mm in at least one of the main diameters (too small of a size makes it difficult to distinguish hyperintense rim from the hypointense core on diffusion and to measure the encephalomalacic area on follow-up); (2) absent follow-up scan or less than 2 out of 3 sequences available; and (3) patients with other indication for laser ablation such as focal cortical dysplasias (FCD).

The MRI protocol is detailed in Table 1.

To measure the area of signal change, we used the product of the two maximum perpendicular diameters drawn in the axial plane; this is a reproducible and standardized method used to measure pediatric tumors [7] and is considered an excellent surrogate of tumor volume without the problems of reproducibility, time demands and inter-reader variability that are associated with volumetric measurements [8,9,10].

We assessed the area of signal change on T2WI, DWI (both rim of diffusion restriction and internal hypointense core) and rim of contrast enhancement on T1WI as a percentage of the area of the whole hamartoma and compared the results with the encephalomalacia (fluid area on T2WI on follow-up scan) as a percentage of the whole hamartoma on follow-up (Figure 1). Where residual disconnected tissue was present within the encephalomalacic area, it was considered together with it, as the disconnected tissue is no longer active (Figure 2).

### 3.2. Statistical Methods

Continuous parametric variables are presented as mean ± standard deviation.

Normal distribution was assessed using Shapiro–Wilk test (*p* > 0.05) and histogram plots.

One-way repeated measures ANOVA was used to determine whether the mean difference between observations is statistically significant. In detail: the percentage of malacic damage to the residual lesion compared to the percentage of intraoperative damage visualized intraoperatively by the different sequences. *p* values < 0.05 were considered statistically significant. IBM SPSS Statistics 22 (IBM Corp., Armonk, NY, USA) was used to perform the analyses.

## 4. Results

A total of 20 patients met the inclusion criteria (female: 45%). The remaining 19 subjects were excluded for the following reasons: 7 were FCDs, 2 were previously operated making the measurements on follow-up difficult, 4 did not have a follow-up scan, 5 HH were too small, and in 1 the catheter was placed outside the hamartoma.

The mean age at the time of the laser ablation was 5.8 years (±2.5), and the mean time at follow-up was 16 months (±8) (Table 2).

Of note, the T2-weighted intraoperative scan was not available for three patients.

The percentage of intraoperative ablation compared to initial HH size was 101.4 ± 37.7% in T2WI, 93.4 ± 29.2% in post-contrast T1WI and 31.0 ± 11.9% and 87.2 ± 34.8% in DWI images (rim of restriction excluded and included, respectively). The percentage of malacic damage to the residual lesion on follow-up MRI was 38.9 ± 18.4%.

The one-way ANOVA for repeated measures and the post hoc analysis with a Bonferroni adjustment revealed that the difference between the percentage of malacic damage and the percentage of DWI core intraoperative scan was not significant (mean difference 7%, 95% CI −1%–17%, *p* = 0.104). Significant difference was found instead between the percentage of malacic damage and the percentage of rim (or edge) of restriction on DWI intraoperative scan (mean difference, −48.2%, 95% CI −68.3%–−28.1%, *p* < 0.001). Similarly, the percentage of malacic damage in follow-up MRI was lower compared to the percentage of intraoperative ablation in post-gadolinium scan (mean difference −54.4%, 95% CI −67.4%–−41.5%, *p* < 0.001) (Figure 3).

In order to explore the role of intraoperative T2WI, the analysis was repeated including only the 17 cases for which the sequence was available. The analysis showed similar results and demonstrated that the percentage of malacic damage in follow-up MRI was lower compared to the percentage of intraoperative ablation in T2WI (mean difference −60.3%, 95% CI −82.6%–−37.9%, *p* < 0.001).

### Summary of the Results: The Area of DWI Hypointense Core, Immediately after the Ablation, Better Correlates with the Final Encephalomalacic Damage on the Follow-Up Scan

Of note a “core” of hypointensity was sometimes noted on T2WI (Figure 1) but was not visualized in all cases and was much more inhomogeneous than the DWI core, jeopardizing reproducible measurements.

The non-enhancing core in T1WI was also measured and corresponded to the DWI hypointensity core; however, because after contrast is given, it is not possible to repeat the ablation, this sequence is not useful in the intra-operative decision making.

## 5. Discussion

### 5.1. LITT Technique and Thermal Heat Maps

LITT is a minimally invasive technique that over the last decade, has become a procedure of choice for HH treatment, offering an alternative to open craniotomy, with shorter hospital stays, higher rates of seizure control and less morbidity. Unlike radiofrequency ablation and stereotactic radiosurgery, MRgLITT is performed with MRI compatible equipment, which means that with this technique, specific MRI sequences can be performed after each ablation. During the ablation, heat maps show the areas of ongoing damage [11].

The goal of LITT is not to coagulate the entire mass but to disconnect the HH from hypothalamic and mammillothalamic tracts [11] by delivering heat: a temperature range of 50–80 °C transferred to tissue for a few seconds causes protein denaturation and DNA damage, thus inducing necrosis. It is possible to plan laser temperature target threshold areas on images before the ablation, better to spare the surrounding tissues [12].

Those Thermal damage threshold (TDT) areas define the temperature level delivered to the lesion and to the surrounding tissues and can be visualized during the procedure by the use of thermal maps. If the temperature delivered exceeds the planned value, the laser automatically stops [13,14,15].

Visualizing the temperature of a tissue in the form of a thermal map is possible by utilising temperature sensitivity of different MR parameters such as proton density, T1 and T2 relaxation times, diffusion coefficient, magnetization transfer and proton resonance frequency (PRF) [16,17]. Thermal maps are generated by a visualization software subtracting fast-spoiled gradient-recalled phase images after and before administration of thermal energy [15,17]; a mathematical model (function of time and temperature—Arrhenius model) is used to estimate the area of irreversible damaged tissue, and it is shown on the map as a colored area, defined as the thermal damage estimate (TDE) [6,12].

During the procedure, thermal imaging is run repeatedly at short time intervals on preselected slides and planes; it serves as guide to stop the procedure when the target area is shown as an area of irreversible damage or if temperature targets exceed the planned thresholds. Those maps have the advantage of offering a very high time resolution, being dynamic real-time images, but are limited by low spatial resolution, and they can be subject to pulsation and metal artifacts [13,14,18,19].

For this reason, once the ablation is completed, some standard MRI sequences (T2, FLAIR, DWI) can be performed to confirm the extent of ablation and the presence of potential residual hamartoma. A T1 sequence after contrast injection is also performed at the end of the procedure to show the effective damaged tissue [20].

Notwithstanding the good correspondence between T1 post-contrast and the extent of necrotic tissue, this sequence cannot be used to guide intraoperatively the procedure because it is not possible to perform additional ablation after contrast administration.

For this reason, alternative MRI sequences, with high spatial resolution and more anatomical definition than a thermal map, can be an additional useful tool to guide intraoperative decisions on the extent of the ablation, thus helping to avoid over- or under-treatment.

### 5.2. Literature Analysis

To our knowledge, no previous study has correlated the appearance of the lesion immediately after each ablation on standard T2WI and DWI, with immediate post-contrast T1WI and then with the extent of the malacic area in long-term follow-up.

A study from Parisi et al. compared standard MRI sequences (FLAIR, DWI, GRE and post-contrast T1WI) performed 24 h after the procedure to identify the most reliable sequences in assessing the volume of the ablated brain metastatic lesion. They concluded that DWI is the second most reliable sequence after post-contrast T1WI [20]. A study from Patel and colleagues [18] analyzed the variation in volume of intracranial primitive tumors and metastasis treated with LITT. They observed an increase in size of the lesion immediately after the ablation; they also highlighted the difficulty in differentiating true lesion from edema. It has to be considered that the lesion included in their study were primitive brain tumors or metastasis located in different areas than hypothalamic hamartomas; thus, it is possible that a degree of over ablation could be performed without significant damage. They also measured the lesions on T1WI post-gadolinium sequences and reported that there might be an overestimation of tumor volume in the immediate post-procedure due to damage to vascular integrity [21].

Given the established relation between the intraoperative thermal damage estimate (TDE) and post-contrast T1WI post-operative lesion, a study from Munier et al. [22] analyzed the effect of the possible presence of artifacts in the intraoperative thermal map and post-operative imaging: they compared TDE presenting artifact with immediate post-ablation post-contrast T1WI sequence. They concluded that artifacts are common on thermal maps and their presence results in high discordance between TDE and post-operative ablated area [22].

Gadgil et al. [19] analyzed the relation between the residual volume of ablated hypo- thalamic hamartomas and clinical outcome. They found that clinical outcome is correlated more to the disconnection of the lesion from the seizure network than to volume reduction of the lesion. They measured the volume of lesions immediately after ablation, observing a correspondence between DWI and post-contrast T1WI [19]. This finding is in line with our data, as T1WI post-contrast and rim (or edge) of DWI restriction present similar volume; however, our data suggest the additional importance of the DWI core, which shows higher correlation with malacic area on follow-up.

Finally, Salem et al. correlated imaging appearances of ablated brain lesions with histology. They describe a central area immediately surrounding the probe that is hypointense on T2WI and hyperintense on T1WI, which is histologically an area of central necrosis that increases in size after each ablation and a peripheral area that corresponds to irreversible edema, which is hyperintense on T2WI and hypointense on T1WI and enhances due to blood–brain barrier damage. Surrounding the peripheral zone, there is an area of reversible vasogenic edema that can often enhance due to contrast leakage; this can lead to overestimation of the ablated lesion on post-contrast T1WI [15]. The description of two histological different areas as a result of the ablation is in line with our imaging findings, as we observed a central hypointese DWI core, which could correspond to necrosis, and a restricting DWI edge and enhancing rim, which could correspond to edema. However, we found that this peripheral enhancing edematous rim overestimates the area of damage and thus may represent reversible damage.

In summary, we demonstrated that intraoperative sequences, in particular DWI core, can be a helpful tool to check the progressive damage to the hamartoma in real time after each ablation (Figure 4). This is critical in guiding the ablation progression while the goal is to destroy or disconnect as much tissue as possible, it is paramount to avoid damage to sensitive structures such as mammillary bodies, fornices and mamillo-thalamic tracts. T2WI and post-contrast enhancing rim overestimate the final irreversible damage.

While the presumed necrotic damage can be evaluated after each ablation with a high spatial resolution DWI sequence, during the ablation, the damage is evaluated with thermal map (high temporal/low spatial resolution), making the two techniques complementary.

### 5.3. Limitations

A limitation of this study is that it is based on radiological appearances only. In particular, despite being able to include the disconnected tissue in some of the cases (see Figure 2F), we do not know whether the reversible edema, not evolving in encephalomalacia (i.e., rim of DWI restriction and contrast enhancement), translates to functionally disconnected tissue, hence impacting the procedure’s outcome.

Another important limitation is that, due to the retrospective nature of the study, we could not compare side-by-side the MRI findings with the heat maps (which are dynamic maps not recorded on PACS); however, the fact that the maps are visualized on one or two slice only (because of technical limitations) and they have high temporal but relatively low spatial resolution, is strongly supportive of an added role of the intraoperative sequences.

Finally, prospective clinical data are being collected to establish how the radiological findings correlate with clinical results.

## 6. Conclusions

The use of intraoperative MRI sequences during LITT provides better anatomical definition than thermal maps and can be used after each ablation to help define, with high spatial resolution, the irreversibly damaged area. The hypointense core on DWI is the best predictor of the final tissue damage.

## Figures and Tables

**Figure 1 diagnostics-13-02331-f001:**
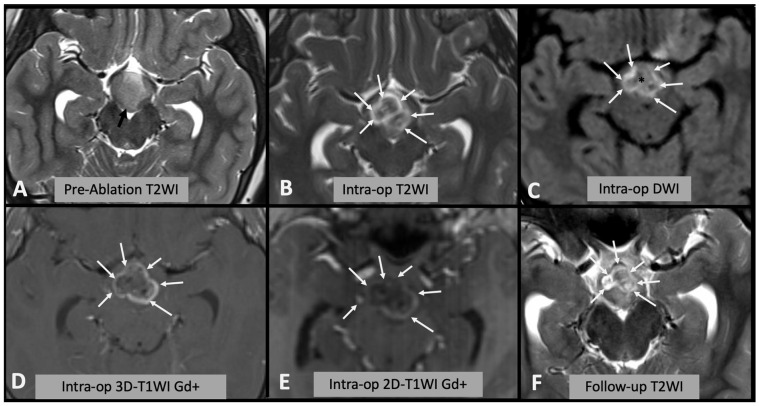
(**A**) Axial T2WI showing the appearances of the hypothalamic hamartoma before laser ablation (black arrow). (**B**) Intra-operative axial T2WI immediately after laser ablation, showing marked T2WI hyperintense oedema within the hamartoma (white arrows). (**C**) Intra-operative axial DWI after laser ablation showing hyperintense rim (white arrows) surrounding a hypointense core (asterisk). (**D**,**E**) Post-contrast T1WI sequences showing enhancing rim within the hamartoma (arrows); after injection of contrast it is not possible to repeat laser ablation; thus, this sequence is not useful for intra-operative decision making. Note the differences between (**D**) spin-echo 2D T1WI and (**E**) 3D “MPRAGE” T1WI; in the latter, the rim is less evident due to technical differences. (**F**) Follow-up axial T2WI showing the encephalomalacic area of permanent damage.

**Figure 2 diagnostics-13-02331-f002:**
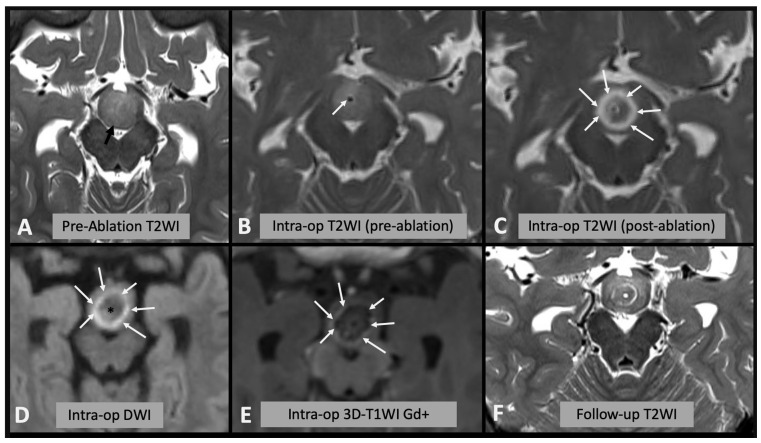
(**A**) Axial T2WI showing the appearances of the hypothalamic hamartoma before laser ablation (black arrow). (**B**) Intra-operative axial T2WI immediately before laser ablation, showing the catheter placed in the center of the hamartoma (white arrow). (**C**) Intra-operative axial T2WI immediately after laser ablation, showing marked T2 hyperintense edema within the hamartoma (white arrows). (**D**) Intra-operative axial DWI after laser ablation showing hyperintense rim (white arrows) surrounding a hypointense core (asterisk). (**E**) Post-contrast T1WI sequence showing enhancing rim within the hamartoma (arrows). (**F**) Follow-up axial T2WI showing the encephalomalacic area of permanent damage; note the central area of T2WI hypointensity corresponding to disconnected hamartomatous tissue.

**Figure 3 diagnostics-13-02331-f003:**
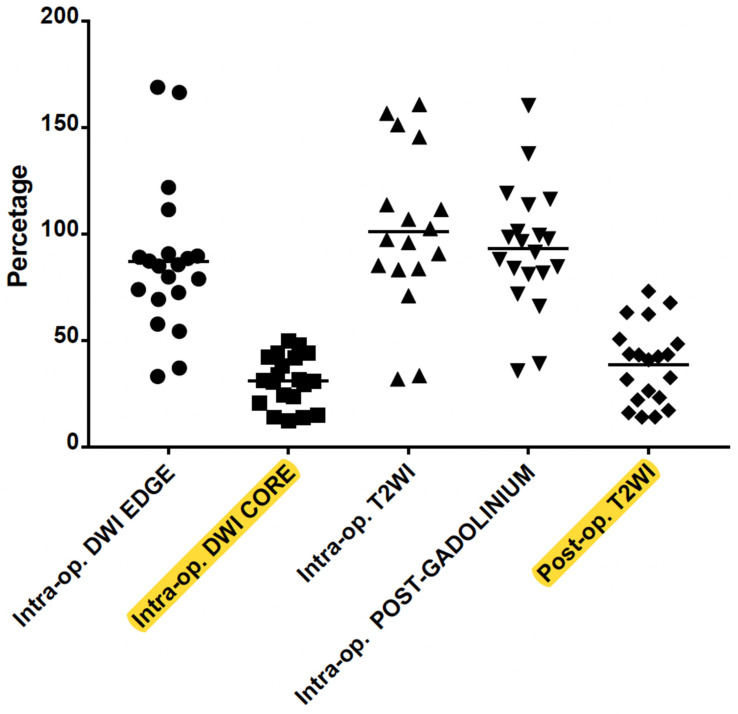
Correlation between intraoperative DWI (hyperintense edge or rim and hypointense core), T2WI edema and post-Gadolinium T1WI rim of enhancement with the post-op encephalomalacia on T2WI. The DWI core is a significant predictor of the final volume of malacic damage in the HH (highlighted in yellow).

**Figure 4 diagnostics-13-02331-f004:**
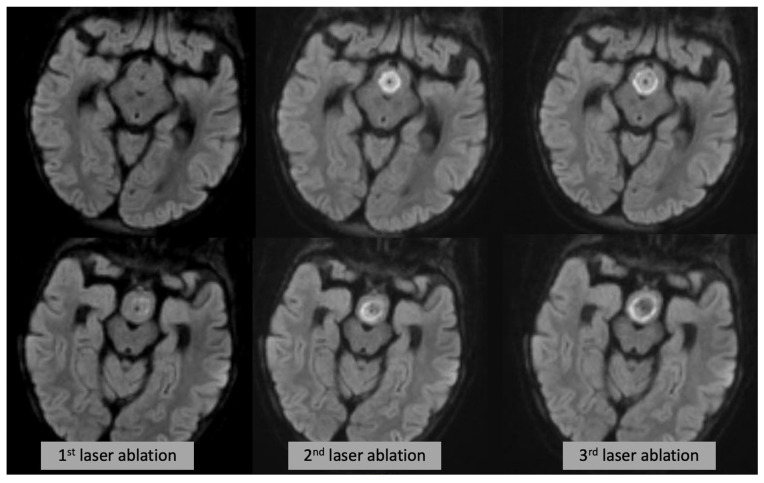
Axial DWI (each row showing a different slice/level) at the time of the first (**left**), second (**middle**) and third (**right**) laser ablation, showing progressive expansion of the DWI changes.

**Table 1 diagnostics-13-02331-t001:** Summary of MRI acquisition prototocols performed at different stages between pre-operative evaluation and follow-up.

Pre-operative evaluation and follow-up (essential sequences):
- 3D T1 MPRAGE, isotropic voxel, 1 mm^3^, with multiparametric reformats in 3 planes - 3D FLAIR, isotropic voxel, 1 mm^3^, with multiparametric reformats in 3 planes - Axial and coronal T2WI 3 mm slice thickness (no gap) - 3D T1WI inversion recovery, isotropic voxel, 1 mm^3^, with multiparametric reformats in 3 planes
Intraoperative evaluation
- Axial T2WI 3 mm slice thickness (no gap) on the hypothalamic region - Axial DWI RESOLVE 3 mm slice thickness (no gap) on the hypothalamic region
Immediate Post-operative evaluation
- Axial T2WI 3 mm slice thickness (no gap) on the hypothalamic region - Axial DWI RESOLVE 3 mm slice thickness (no gap) on the hypothalamic region - post-contrast axial T1 spin-echo 3 mm slice thickness (no gap) - post-contrast 3D T1 SPACE, isotropic voxel, 1 mm^3^, with multiparametric reformats in 3 planes.
Follow-up evaluation
- same protocol as for the pre-operative evaluation.

**Table 2 diagnostics-13-02331-t002:** Product diameters of regions of interest (ROIs) drawn.

Timing	Sequence Contrast, ROI	Mean Product Diameter (mm^2^)	Standard Deviation (mm^2^)
Pre-operative	T2WI HH	241.4	168.5
Intra-operative	DWI (includes edge)	174.8	88.5
Intra-operative	DWI core	71.8	52.8
Intra-operative	T2WI	211.4	109.3
Intra-operative	T1WI post-Gadolinium	195.9	103.3
Post-operative	T2WI encephalomalacia	59.0	39.7
Post-operative	T2WI residual HH	175.8	133.6

## Data Availability

Not applicable.

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
