# Peer review of "Intraoperative MRI Assessment of the Tissue Damage during Laser Ablation of Hypothalamic Hamartoma"

_diagnostics, 2023, doi:10.3390/diagnostics13142331_

Round 1

Reviewer 1 Report

1. The following sentence is a run on and lacks clarity: Among minimally invasive techniques, SRS has the main disadvantage of delayed side effects (up to 3 years after treatment) and a more significant risk of damaging radio- sensitive surrounding structures, like optic chiasm, while with RFA the extent of ablation in a single session is smaller than for Magnetic Resonance guided LITT (MRgLITT) and multiple probe passes with multiple trajectories are needed (4). 

2. There are several paragraphs in the discussion that are only a single sentence long, these should be combined into paragraphs

3. The authors conclude that DWI is better than thermal mapping, however no comparison was performed between DWI and thermal maps, only with T1/T2 sequences so they may not be able to make this conclusion

1. The following sentence is a run on and lacks clarity: Among minimally invasive techniques, SRS has the main disadvantage of delayed side effects (up to 3 years after treatment) and a more significant risk of damaging radio- sensitive surrounding structures, like optic chiasm, while with RFA the extent of ablation in a single session is smaller than for Magnetic Resonance guided LITT (MRgLITT) and multiple probe passes with multiple trajectories are needed (4). 

2. There are several paragraphs in the discussion that are only a single sentence long, these should be combined into paragraphs

Author Response

RESPONSE TO REVIEWER 1

Comments and Suggestions for Authors

  1. The following sentence is a run on and lacks
    clarity: Among minimally invasive techniques, SRS has the main disadvantage of delayed side effects (up to 3 years after treatment) and a more significant risk of damaging radio- sensitive surrounding structures, like optic chiasm, while with RFA the extent of ablation in a single session is smaller than for Magnetic Resonance guided LITT (MRgLITT) and multiple probe passes with multiple trajectories are needed (4)
    .

Thank you for your observation, the sentences has been reformulated in a more complete way as follow:

The treatment of HH consists in the removal or disconnection of the hamartoma from the normal brain, interrupting the seizure network. Currently, besides open surgery, minimally invasive techniques are available, such as stereotactic radiosurgery (SRS), radiofrequency ablation (RFA) and laser interstitial thermal therapy (LITT) (2–4).

Stereotactic radiosurgery and radiofrequency ablation have some limitations compared to laser interstitial thermal therapy.

SRS has the main disadvantage of requiring up to 3 years to demonstrate the efficacy of the treatment and is also associated with a more significant risk of damaging radio-sensitive surrounding structures like optic chiasm.

RFA allows ablating a more limited area in a single session than it is possible to do with LITT, and it also requires the use of multiple probes with different trajectories (4).

MRgLITT is a minimally invasive technique that developed over the last decade, offering an alternative to open craniotomy, with shorter hospital stay, high rate of seizure control and reduced morbidity, becoming the first-choice procedure for HH treatment.

Compared with radiofrequency ablation and stereotactic radiosurgery, MRgLITT has the advantage of being performed with ad MRI compatible equipment, thus allowing a real time assessment of the tissue ablation in real time during the procedure (1,2,4).

  1. There are several paragraphs in the discussion that are only a single sentence long, these should be combined into paragraphs

Thank you for the observation. The sentences have been rephrase as suggested.

  1. The authors conclude that DWI is better than thermal mapping, however no comparison was performed between DWI and thermal maps, only with T1/T2 sequences so they may not be able to make this conclusion

Thank you for pointing out this important consideration.

Thermal maps are not like standard MRI sequences, they are dynamic images, displayed during the procedure to guide it in real time. It is also to be considered that they are a couple of single slices and therefore do not demonstrate the volumetric extent of the ablation. So it is not possible to use them to make a quantitative comparison with standard sequences.

The aim of the study is to add an intraoperative fast MRI sequence that can offer higher spatial resolution to assess with more detail the extent of the ablation, indeed DWI offers improved spatial resolution while the thermal maps offers high temporal resolution, so they should be considered rather complementary. We make this clear in the manuscript.

Indeed, the role of thermal map is to guide the procedure during the ablation (i.e. when to stop) , while the role of DWI is to help to decide whether to proceed with further ablation depending on the extension of the signal change.                                                                  

Comments on the Quality of English Language

  1. The following sentence is a run on and lacks
    clarity: Among minimally invasive techniques, SRS has the main disadvantage of delayed side effects (up to 3 years after treatment) and a more significant risk of damaging radio- sensitive surrounding structures, like optic chiasm, while with RFA the extent of ablation in a single session is smaller than for Magnetic Resonance guided LITT (MRgLITT) and multiple probe passes with multiple trajectories are needed (4).

See answer to point 1 above.

  1. There are several paragraphs in the discussion that are only a single sentence long, these should be combined into paragraphs

RESPONSE TO REVIEWER 3                                                 

Comments and Suggestions for Authors

Overall this is an original study that has clinical implications and worthy of publication I have one suggestion

  1. Please provide seizure free rates of this cohort and compare seizure outcomes to images of size of ablation.

Thank you for your comment. We agree that seizure freedom rates are important to report, this will form part of a second, ongoing, paper, for which prospective data are still being collected.

Reviewer 2 Report

Very good work!

Author Response

(The authors gave the same response as above.)

Reviewer 3 Report

Overall this is an original study that has clinical implications and worthy of publication  I have one suggestion

1. Please provide seizure free rates of this cohort and compare seizure outcomes to images of size of ablation.  

Author Response

(The authors gave the same response as above.)
